# Qualitative and Quantitative Analysis of Ejiao-Related Animal Gelatins through Peptide Markers Using LC-QTOF-MS/MS and Scheduled Multiple Reaction Monitoring (MRM) by LC-QQQ-MS/MS

**DOI:** 10.3390/molecules27144643

**Published:** 2022-07-21

**Authors:** Wen-Jie Wu, Li-Feng Li, Hau-Yee Fung, Hui-Yuan Cheng, Hau-Yee Kong, Tin-Long Wong, Quan-Wei Zhang, Man Liu, Wan-Rong Bao, Chu-Ying Huo, Shangwei Guo, Haibin Liu, Xiangshan Zhou, Deng-Feng Gao, Quan-Bin Han

**Affiliations:** 1School of Chinese Medicine, Hong Kong Baptist University, 7 Baptist University Road, Kowloon Tong, Hong Kong 999077, China; 18482767@life.hkbu.edu.hk (W.-J.W.); 16483294@life.hkbu.edu.hk (L.-F.L.); 11018860@life.hkbu.edu.hk (H.-Y.F.); hycheng10@163.com (H.-Y.C.); 16223551@life.hkbu.edu.hk (H.-Y.K.); 15485021@life.hkbu.edu.hk (T.-L.W.); 18482422@life.hkbu.edu.hk (Q.-W.Z.); liuman@hkbu.edu.hk (M.L.); 16483502@life.hkbu.edu.hk (W.-R.B.); 20481969@life.hkbu.edu.hk (C.-Y.H.); 2Hong Kong Authentication Centre of Valuable Chinese Medicines, Hong Kong 999077, China; gaodf@dongeejiao.com; 3Shandong Technology Innovation Center of Gelatin-Based Traditional Chinese Medicine, Dong-E-E-Jiao Co., Ltd., No. 78, E-Jiao Street, Done-E Country, Liaocheng 252200, China; guosw@dongeejiao.com (S.G.); liuhaibin@dongeejiao.com (H.L.); 4China Resources Biopharmaceutical Co., Ltd., Beijing 100000, China

**Keywords:** authentication, Ejiao, gelatin, peptide marker, database-independent strategy, de novo sequencing

## Abstract

Donkey-hide gelatin, also called Ejiao (colla corii asini), is commonly used as a food health supplement and valuable Chinese medicine. Its growing popular demand and short supply make it a target for fraud, and many other animal gelatins can be found as adulterants. Authentication remains a quality concern. Peptide markers were developed by searching the protein database. However, donkeys and horses share the same database, and there is no specific marker for donkeys. Here, solutions are sought following a database-independent strategy. The peptide profiles of authentic samples of different animal gelatins were compared using LC-QTOF-MS/MS. Fourteen specific markers, including four donkey-specific, one horse-specific, three cattle-specific, and six pig-specific peptides, were successfully found. As these donkey-specific peptides are not included in the current proteomics database, their sequences were determined by de novo sequencing. A quantitative LC-QQQ multiple reaction monitoring (MRM) method was further developed to achieve highly sensitive and selective analysis. The specificity and applicability of these markers were confirmed by testing multiple authentic samples and 110 batches of commercial Ejiao products, 57 of which were found to be unqualified. These results suggest that these markers are specific and accurate for authentication purposes.

## 1. Introduction

Ejiao (colla corii asini) is a gelatin made from donkey (*Equus asinus*) skin. It is popularly used as a food health supplement and valuable Chinese medicine [1,2]. Ejiao exerts diverse beneficial effects, such as optimizing the immune response, nourishing the blood, and delaying senescence [3,4,5,6,7]. Ejiao market sales reached more than USD 6 billion in 2020. However, because the demand is increasing much faster than the supply, various adulterants, particularly gelatins made from cattle, pig, and horse hides, are frequently found in the market [8].

Peptide markers have been developed for authentication purposes. For example, in the Chinese Pharmacopoeia, three peptides are used as authentication markers for Ejiao [2]. Li et al. found two peptide markers for Ejiao by an improved proteomics approach, combining bioinformatics-based prediction [8]. Liu et al. found two peptide markers for deer-hide gelatin by combining set theory analysis. The result suggested that bioinformatics-based prediction might ignore posttranslational modifications that occur during the extraction procedure, which cause a false positive [9].

The specificity of these markers relies on the proteomics database (Figure 1A) [2,8,9,10,11,12,13,14,15], but many of the databases are shared by different species, e.g., that shared by donkey (*Equus asinus*) and horse (*Equus caballus*) [16,17,18,19]. Kumazawa et al. identified a total of 396 peptides for deer glue, including 55 peptides from bovine (*Bos taurus*) and 37 peptides from sheep (*Ovis aries*) [16]. The myoglobin peptide HPGDFGADAQGAMTK is horse-specific in the UniProt database [20]; however, it also exists in donkeys [18]. Furthermore, Prandi et al. found that all the peptides identified in horse meat were also present in donkey meat. Therefore, this myoglobin peptide has to be used as a marker peptide for equine meat (horse and donkey) in their study [18]. The specific peptide GPPGAAGPPGPR for authentication of Ejiao in the Chinese Pharmacopoeia is also found in horse-hide gelatin [2,8]. More importantly, many peptides are not recorded in these protein databases [16,18]. For example, information on pig COL1A1 is lacking [8]. The deer type I collagen sequence is not included in public databases [16]. As a result, there are no donkey-specific markers. New reliable and specific marker(s) must be found in another way (Figure 1B).

This study first directly compared the peptide profiles among donkey-hide and other animal-hide gelatins. We prepared 11 homemade animal-hide gelatin samples and collected 4 batches of reference standard materials and 110 commercial Ejiao samples. Species-specific markers were found using LC-QTOF-MS/MS, after comparing base peak chromatographs (BPC) and extracted ion chromatograms (EIC) of these samples. Furthermore, a typical specific peptide marker for each animal species was selected to develop a quantitative LC-QQQ multiple reaction monitoring (MRM) method, enabling highly sensitive and selective analysis. These markers were proven specific and reliable for qualitative and quantitative quality analysis.

## 2. Results and Discussion

### 2.1. Selection of Species-Specific Peptide Markers

There were several steps to find authentication markers. Firstly, as shown in Figure 2, we scanned every ion in the range of *m*/*z* 100–1200 and found 569 ions in standard donkey-hide gelatin (Appendix A). Similarly, we screened every ion in standard horse gelatin, cattle gelatin, and pig gelatin, and found 575, 474, and 453 ions, respectively (Appendix A).

Secondly, we further screened these 569 donkey ions in other animal samples using the extracted ion chromatogram (EIC) mode (Figure 3). In nature, due to the presence of ^13^C and ^2^H, the molar mass of the most abundant isotope is increased by 1 Da. Thus, its corresponding mass–charge ratio (*m*/*z*) is increased by 1/z, thereby confirming the charge number. Ions can be largely confirmed as peptides if they are doubly, triply, or quadruply charged peaks. Because the isotope with *m*/*z* = 1150.0366 (two charges, Appendix A) is another form (*m*/*z* = 767.0313, Figure 5A) of the marker (*m*/*z* = 766.6952, DM4, Figure 3), it was not regarded as new. Our results showed 421 ions in all four gelatins, 80 ions in both donkey and horse gelatins but not in cattle or pig gelatin (Appendix A), and only 7 donkey-specific ions (Table 1).

All the other gelatins were screened for the 575 ions from horse gelatin, 474 ions from cattle gelatin, and 453 ions from pig gelatin. Finally, four horse-specific, three cattle-specific, and six pig-specific ions were found (Appendix A). These findings were verified in multiple batches of reference samples (Figure 3). Three horse-specific marker candidates (*m*/*z* 455.2539, 552.7825, and 716.4201) were found in only one batch of horsehide, suggesting low consistency; therefore, they were not considered as markers (Figure 3).

In summary, 17 specific marker ions (Table 1), namely 7 donkey-specific (DM1~DM7), 1 horse-specific (HM1), 3 cattle-specific (CM1~CM3), and 6 pig-specific marker ions (PM1~PM6), were successfully found.

### 2.2. Comparison with Reported Markers

We also checked the specificity of the reported donkey-specific markers (Table 2). There are 32 reported donkey-specific markers in total [21,22,23,24,25,26,27]. We found 20 of them in donkey and horse gelatins, and the remaining 12 markers showed too low intensities to be detected in any animal gelatins. It is worth noting that three Ejiao marker peptides recommended in the Chinese Pharmacopeia (GPAGPTGPVGK, *m*/*z* 469.25; GPPGAAGPGPR, *m*/*z* 539.8; GEAGAAGPAGPAGPR, *m*/*z* 618.35) were all found in horse-hide gelatin. These reported Ejiao authentication markers are not specific enough. Because horses and donkeys share protein databases under the name of Equus, that is, there is no donkey-specific protein database.

The specificity of the markers mentioned above is limited by the database. The precondition of these markers’ specificity is that the database is powerful enough. However, the fact is that these databases are growing and keep being updated. If the database fails to provide sufficient/updated information, the specificity of selected markers will be significantly weakened. In this study, we also tried the conventional strategy: to search data against Bos taurus (cattle), Sus scrofa (Pig), and Equus (donkey and horse) protein databases downloaded from UniProt KB (8 January 2022), and select those matched peptides having 99% confidence. Then, we obtained six so-called donkey-specific markers; however, they all existed in horse gelatin samples (Table 3). This failure highlights the limitation of the existing databases.

### 2.3. De Novo Sequencing Identification of the Newly Found Donkey-Specific Markers

De novo sequencing was used to determine the seven new donkey-specific marker ions’ sequences. Mass spectrometry raw data of target peptide profiles were submitted to carry out de novo sequencing in PEAKS Studio using a tolerance of precursor and fragment mass of 0.1 Da. De novo peptides, whose average local confidence (ALC) scores with more than 50%were selected. Four out of the seven marker ions are matched with peptides generated by missed trypsin cleavage, including DM1, KCSLDYGKDHEPVQVGPR; DM2, CFWKYNGLPGSAFCFDK; DM3, NPTWNKPKPAYGHAGVGSMK; DM4, RTMWPFFEALCGGFGASNHK (Figure 4). ALC scores are 55%, 83%, 61%, and 71% respectively. Considering that the four donkey-specific marker peptides have 18, 17, 20, and 20 amino acids, respectively, it is difficult to match the amino acid sequences completely. Thus, the sequences of the four peptides serve as a reference.

### 2.4. Quantitative Analysis of Commercial Ejiao Products

A quantitative LC-QQQ multiple reaction monitoring (MRM) method was further developed. (I) The MRM transitions were selected based on MS/MS spectra from LC-QTOF-MS/MS (Figure 5 and Appendix A). (II) The collision energy (CE) of the MRM transitions was optimized for sensitivity from 3 to 55 Volts and the CE with the highest response was selected (Table 4). (III) The specificity of the nine peptide markers was confirmed by comparing authentic samples. (IV) The linearity was plotted, and there were good linear relationships between the peak areas of these peptide markers and the content of the corresponding animal gelatins, especially Ejiao (Appendix A). (V) Based on the established linearity, the method validation result showed that the proposed method has good intra-day precision, inter-day precision, and spike recovery (Table 5). (VI) The proposed method was applied for quantitative analysis of all commercial Ejiao products (Table 6).

One marker with a good linear relationship and high sensitivity was selected for each species (DM4 for donkey, HM1 for the horse, CM2 for cattle, and PM2 for pig) and used to quantify 110 commercial Ejiao samples (Appendix A). MS/MS information of these markers was provided to confirm the ions (Figure 5). The quantitative analysis results were highly consistent with markers in the Chinese Pharmacopoeia [2]. As summarized in Table 6 and Appendix A, 57 out of the 110 samples (51.8%) were fake, including 45 cattle gelatin, 5 horse gelatin, 3 pig/horse gelatin, and 1 cattle/horse gelatin products. Donkey gelatin and horse gelatin were simultaneously found in two products, and another did not contain any gelatins. These results suggest the new markers are reliable.

Our method seems to be more powerful than the method proposed in the Chinese Pharmacopoeia. As shown in Table 6, there are controversial results for some samples containing horse-hide gelatin because markers in the Chinese Pharmacopoeia are donkey–horse shared markers, generating false-positive results when a sample contains horse-hide gelatin. Even when subtracting the corresponding content from horse-hide gelatin, the authentication results based on markers in the Chinese Pharmacopoeia are not ideal because the relative content of donkey markers in the Chinese Pharmacopoeia and horse-specific markers are different in different samples containing horse-hide gelatin (Figure 6). By contrast, our new markers directly indicate the presence of donkey-hide gelatin without the need for any other test or calculation.

Interestingly, the authentication results showed that there were close relationships between authenticity and price, registration, and manufacturers. Among these tested 110 samples, the cost of all the authentic samples was above HKD 2450/500 g, while the fake samples were all priced below HKD 2250/500 g (Appendix A). Seventy-four samples showed the manufacturer’s name; of them, 53 (72%) were authentic, while the other 36 samples, without the manufacturer’s name, were proved to be fake. Forty-six samples were labeled with a National Medical Products Administration (NMPA) License No. and 64 samples lacked this label. It is impressive that 45 out of 46 licensed products were authentic, while only 8 samples were proved to be authentic among 64 non-licensed products (Appendix A). These data suggest that there is considerable confusion in the Ejiao market. The data also indicate that high prices and the presence of an NMPA license number tend to be predictors of authenticity. More accurate and reliable authentication methods are greatly desired.

## 3. Materials and Methods

### 3.1. Chemicals and Materials

Trypsin (sequencing grade) was bought from Shanghai Yuanye Bio-Technology Co., Ltd. (Shanghai, China). Ammonium bicarbonate (NH_4_HCO_3_) and formic acid (FA) were purchased from Sigma Aldrich (St. Louis, MO, USA). LC-MS-grade acetonitrile and methanol were provided by RCI Labscan Limited (Bangkok, Thailand). Water used was purified with a Millipore Milli-Q water purification system. Shandong Dong-E E-Jiao Co., LTD (Shandong, China) provided homemade gelatins, including two donkey-hide gelatins, three horse-hide gelatins, three pig-hide gelatins, and three cattle-hide gelatins. Standard materials of two donkey-hide gelatins (Lot No. 121274-201703; 121274-201202), one pig-hide gelatin (Lot No. 121745-201701), and one cattle-hide gelatin (Lot No. 121695-201802) were obtained from the National Institutes for Food and Drug Control (Beijing, China). 110 commercial Ejiao samples were bought in the Hong Kong market.

### 3.2. Sample Preparation

The sample pretreatment process was based on our previous study with minor modifications [28]. The powder (5 mg) of animal gelatins was dissolved in 500 μL 1% ammonium bicarbonate (ABC) and sonicated for 30 min. Then 200 μL of trypsin (5 mg/mL) was added (trypsin–sample = 1:5). The mixture was allowed to digest at 37 °C for 18 h. After digestion, 20 μL of formic acid was added to terminate digestion with a final pH of less than 4. After that, 400 μL of the solution was mixed with 1 mL of methanol in a 1.5 mL EP-tube to make a 70% MeOH solution. Subsequently, the solution was centrifuged at 15,000 rpm for 15 min. The supernatant was collected for further analysis.

### 3.3. LC-QTOF-MS/MS Analysis

The separation was performed on an Agilent 1290 UHPLC system (Agilent Technologies, Santa Clara, CA, USA) equipped with a binary pump, a thermostatic column, an auto-sampler, a degasser, and a diode-array detector. The system was controlled by Mass Hunter B.06 software. An ACQUITY UPLC BEH C18 (2.1 mm × 100 mm, 1.7 μm, Waters, Milford, CT, USA) chromatographic column was used and eluted with a linear gradient of 0.1% formic acid in water (A) and 0.1% formic acid in acetonitrile (B) at a flow rate of 0.35 mL/min and at a temperature of 40 °C. The solvent gradient programming was as follows: 0–5 min, 1% B; 5–30 min, 1–25% B; 30–32 min, 25–75% B; 32–33 min, 75–100% B; 33–34.1 min, 100–1% B; 34.1–38 min, 1% B. The injection volume was 2 μL.

MS data were collected from an Agilent 6540 Q-TOF mass spectrometer (Agilent Technologies, Santa Clara, CA, USA) equipped with a quadrupole-time-of-flight (Q-TOF) mass spectrometer and a JetStream electrospray ion (ESI) source. Data acquisition was controlled by MassHunter B.06 software (Agilent Technologies). The optimized operating parameters in the positive ion mode were as follows: nebulizing gas (N2) flow rate, 7.0 L/min; nebulizing gas temperature, 300 °C; JetStream gas flow, 8 L/min; sheath gas temperature, 350 °C; nebulizer, 40 psi; capillary, 3000 V; skimmer, 65 V; Oct RFV, 600 V; fragmentor voltage, 150 V. The mass scan range was set as mass to charge (*m*/*z*) 100–2000. MS/MS produces parallel alternating scans that provide precursor ion information at low collision energy. In contrast, MS/MS produces full scans that provide information about fragment masses, precursor ions, and neutral loss at high collision energy. The collision energies for Auto MS/MS analysis were 20 V and 40 V.

### 3.4. LC-QQQ-MS/MS Analysis

The separation was performed on an Agilent 1290 UHPLC system (Agilent Technologies, Santa Clara, CA, USA) equipped with a binary pump, a thermostatic column, an auto-sampler, and a degasser. The system was controlled by Mass Hunter B.06 software. An ACQUITY UPLC BEH C18 (2.1 mm × 100 mm, 1.7 μm, Waters, Milford, CT, USA) chromatographic column was used and eluted with a linear gradient of 0.1% formic acid in water (A) and 0.1% formic acid in acetonitrile (B) at a flow rate of 0.35 mL/min and a temperature of 40 °C. The solvent gradient programming was as follows: 0–12 min, 5–35% B; 12–14 min, 35–100% B; 14–16 min, 100% B; 16–16.1 min, 100–5% B; 16.1–18.5 min, 5% B. The injection volume was 2 μL.

MS data were collected from an Agilent 6460 QQQ mass spectrometer (Agilent Technologies, Santa Clara, CA, USA) equipped with a triple quadrupole (QQQ) mass spectrometer and a JetStream electrospray ion (ESI) source. Data acquisition was controlled by MassHunter B.06 software (Agilent Technologies). The optimized operating parameters in the positive ion mode were as follows: nebulizing gas (N2) flow rate, 8.0 L/min; nebulizing gas temperature, 300 °C; JetStream gas flow, 8 L/min; sheath gas temperature, 350 °C; nebulizer, 45 psi; capillary, 3500 V.

## 4. Conclusions

Fourteen specific markers were found to authentic different Ejiao-related gelatins, after comparing their peptide profiles generated by trypsin digestion. Four donkey-specific markers were identified as new peptide markers by de novo sequencing. An accurate quantitative analysis method was established based on these specific markers with sufficient validation data. These new markers showed higher specificity and reliability, and successfully revealed 57 fakes out of 110 commercial Ejiao samples.

## Figures and Tables

**Figure 1 molecules-27-04643-f001:**
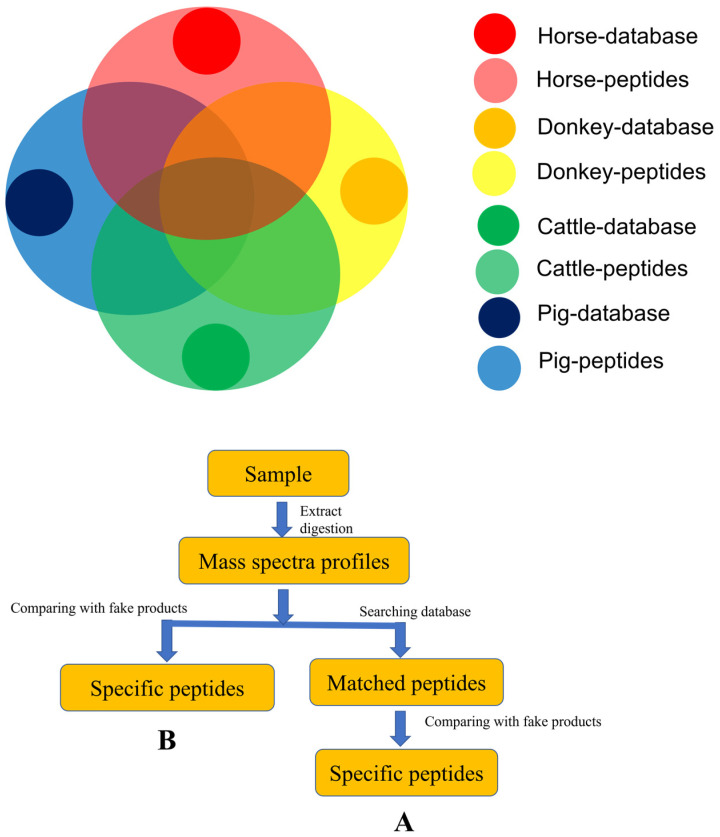
Strategies developed for screening specific peptides. (**A**) Proteomic strategy; (**B**) our strategy. Our strategy (**B**) directly finds specific peptides by comparing the peptide profiles between authentic and fake samples. In contrast, conventional proteomic strategy (**A**) finds specific peptides by searching the related protein database.

**Figure 2 molecules-27-04643-f002:**
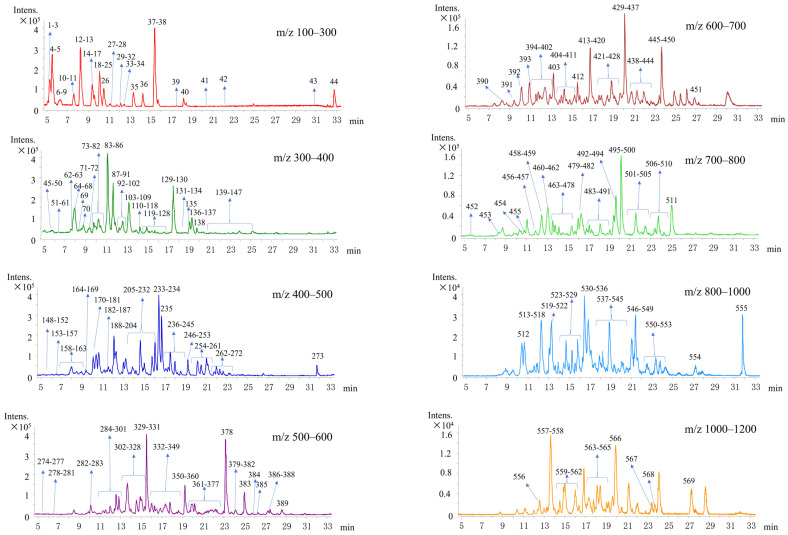
Selected ions from LC-Q-TOF-MS base peak chromatograms (BPC) in the scan range of *m*/*z* 100–1200 of the trypsin hydrolysates of the reference material of donkey skin gelatin.

**Figure 3 molecules-27-04643-f003:**
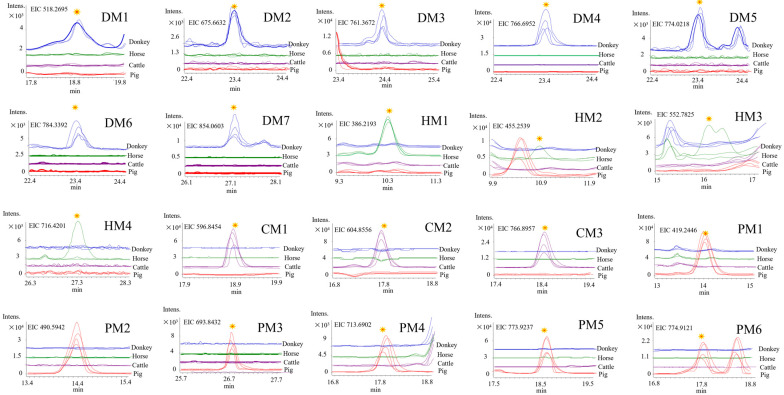
Extracted ion chromatograms of specific marker ions for four animal gelatins. The mass accuracy is 0.1 Da.

**Figure 4 molecules-27-04643-f004:**
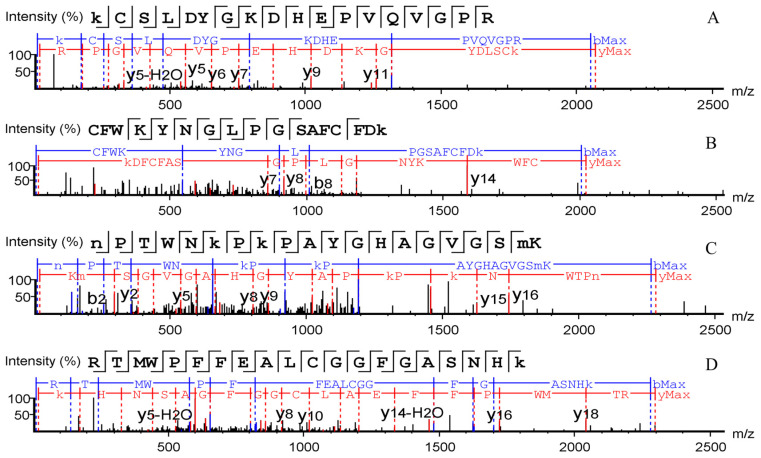
The sequence of four donkey-specific marker peptides identified by de novo sequencing (in PEAKS Studio software using a tolerance of precursor and fragment mass of 0.1 Da), including (**A**) DM1, KCSLDYGKDHEPVQVGPR; (**B**) DM2, CFWKYNGLPGSAFCFDK; (**C**) DM3, NPTWNKPKPAYGHAGVGSMK; (**D**) DM4, RTMWPFFEALCGGFGASNHK.

**Figure 5 molecules-27-04643-f005:**
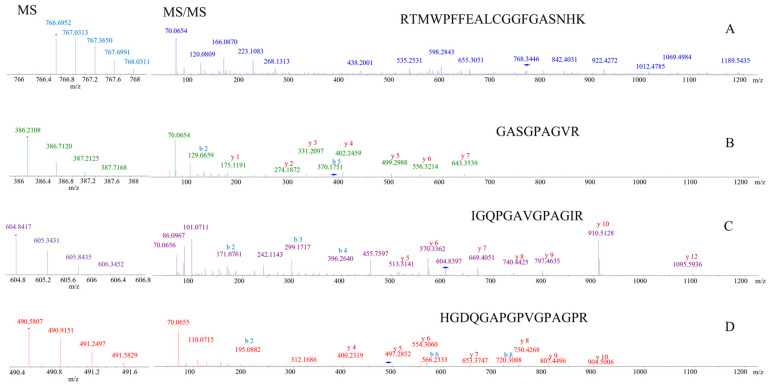
MS and MS/MS spectra of representative high-sensitivity marker for each animal species, including (**A**) donkey-specific marker 4, (**B**) horse-specific marker 1, (**C**) cattle-specific marker 2 and (**D**) pig-specific marker 2. Peptide sequences of (**B**–**D**) are obtained by searching protein databases. By comparing with Skyline software, their matched b and y ions are indicated.

**Figure 6 molecules-27-04643-f006:**
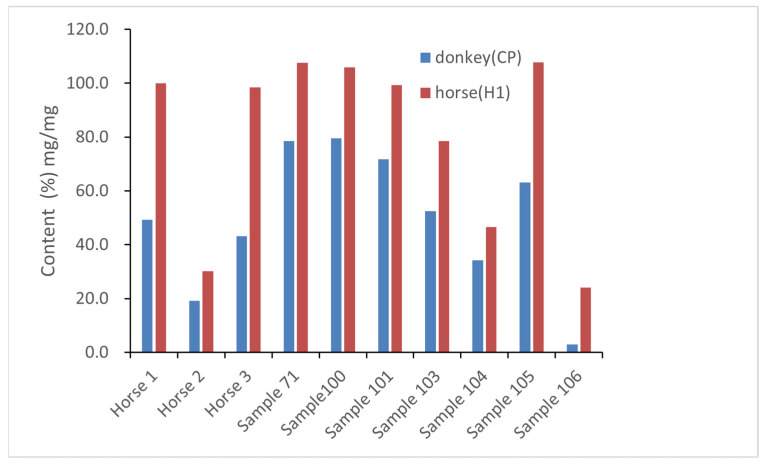
Comparison of relative content of donkey marker in the Chinese Pharmacopoeia (**blue**) and horse-specific marker 1 (**red**) in pure horse-hide gelatins. Horse 1, 2, and 3 are homemade horse-hide gelatins; other samples are commercial Ejiao samples mentioned above.

**Table 1 molecules-27-04643-t001:** The specific marker ions found in the peptide fragments produced from trypsin digestion of different animal skin gelatins by LC-Q-TOF-MS analysis.

	Marker Ion ID	*m*/*z*	Retention Time (min)	Charge ^a^	Donkey	Horse	Cattle	Pig
Donkey	DM1	518.2695	18.92	4	✚ ^b^	- ^b^	-	-
	DM2	675.6632	23.36	3	✚	-	-	-
	DM3	761.3672	24.24	3	✚	-	-	-
	DM4	766.6952	23.36	3	✚	-	-	-
	DM5	774.0218	23.36	3	✚	-	-	-
	DM6	784.3392	23.36	3	✚	-	-	-
	DM7	854.0603	27.08	3	✚	-	-	-
Horse	HM1	386.2108	10.36	2	-	✚	-	-
Cattle	CM1	596.8454	18.90	2	-	-	✚	-
	CM2	604.8556	17.84	2	-	-	✚	-
	CM3	766.8957	18.34	2	-	-	✚	-
Pig	PM1	419.2446	14.04	2	-	-	-	✚
	PM2	490.5942	14.40	3	-	-	-	✚
	PM3	693.8432	26.70	2	-	-	-	✚
	PM4	713.6902	17.79	3	-	-	-	✚
	PM5	773.9237	18.51	2	-	-	-	✚
	PM6	774.9121	17.79	2	-	-	-	✚

^a^ These marker ions showed two/three/four charges and were deduced to be peptides. ^b^ “✚” indicates the existence of the individual markers. “-” means that the specific marker was not detected. The mass accuracy was set as 0.1 Da.

**Table 2 molecules-27-04643-t002:** Specificity evaluation of the published peptide markers of donkey-hide gelatin by LC-Q-TOF-MS analysis.

No.	*m*/*z*	Charge	Donkey	Horse	Cattle	Pig	References
1	393.2	2	✚ ^c^	✚	✚	- ^c^	[24]
2	469.244	2	✚	✚	-	-	[25]
3	469.25 ^a^	2	✚	✚	-	-	[2]
4	523.2746	2	✚	✚	✚	✚	[1]
5	539.774	2	✚	✚	-	-	[25]
6	539.8 ^a^	2	✚	✚	-	-	[2]
7	570.2882	2	✚	✚	✚	✚	[26]
8	570.2891	2	✚	✚	✚	✚	[8]
9	618.35 ^a^	2	✚	✚	-	-	[2]
10	618.795	2	✚	✚	-	-	[25]
11	631.8045	2	-	-	-	-	[1]
12	649.3408	3	✚	✚	-	-	[22]
13	660.3151	2	-	-	-	-	[1]
14	661.5864	4	-	-	-	-	[22]
15	664.8349	2	✚	✚	✚	✚	[1]
16	680.3351	2	-	-	-	-	[1]
17	690.6957	3	-	-	-	-	[22]
18	724.8451	2	-	-	-	-	[22]
19	733.3581	2	✚	✚	-	-	[22]
20	751.3628	2	-	-	-	-	[22]
21	765.8	2	-	-	-	-	[21]
22	765.8556	2	✚	✚	-	-	[23]
23	765.867	2	✚	✚	-	-	[25]
24	765.9142	2	✚	✚	-	-	[22]
25	766.4	2	✚	✚	-	-	[27]
26	767.7234 ^b^	3	-	-	-	-	[22]
27	802.9325	2	✚	✚	-	-	[22]
28	806.1683	4	-	-	-	-	[22]
29	902.4576	2	✚	✚	-	-	[22]
30	910.4554	2	-	-	-	-	[22]
31	921.4649	2	✚	✚	-	-	[22]
32	1073.08	2	-	-	-	-	[22]

^a^ The marker peptide is from Chinese Pharmacopeia (2020 edition). ^b^ The marker (*m*/*z* 767.7234) has a low intensity in donkey-hide gelatin (10^3^). Moreover, it failed to be distinguished from the marker’s isotope (*m*/*z* = 767.6991, Figure 5A) (*m*/*z* = 766.6952, DM4) due to mass accuracy. ^c^ “✚” indicates the existence of the individual markers. “-” means that the specific marker was not detected. The mass accuracy was set as 0.1 Da. A marker of the animal gelatin confirmed must satisfy the following criteria: (1) the specific peak was found; (2) the specific peak shows a high intensity (≥10^4^).

**Table 3 molecules-27-04643-t003:** Specificity evaluation of 6 markers found in authentic donkey sample by searching protein databases. All were also found in the authentic horse sample.

No. ^a^	*m*/*z*	RT (min)	Charge	Horse	Cattle	Pig
1	427.7377	10.96	2	✚	-	-
2	434.7462	10.56	2	✚	-	-
3	469.2702	13.12	2	✚	-	-
4	492.5830	12.15	3	✚	-	-
5	552.6176	12.69	3	✚	-	-
6	746.8774	13.75	3	✚	-	-

^a^ After LC-Q-TOF-MS/MS analysis, the corresponding original data were searched against *Bos taurus* (cattle), *Sus scrofa* (Pig), and *Equus* (donkey and horse) protein databases downloaded from UniProt KB (8 January 2022). By searching protein databases, peptides with 99% confidence, which means that 99 correct peptides and 1 incorrect, were selected. Then, by comparing these high-confidence peptides from four animal skins, donkey-specific markers were screened. These makers ranked on the high-confidence peptide list of donkey skin, but not in that of horse, cattle, or pig skin. “✚” indicates the existence of the individual markers. “-” means that the specific marker was not detected.

**Table 4 molecules-27-04643-t004:** The precursor ion, retention time, product ion, and collision energy of the selected species-specific markers.

No. ^a^	Precursor Ion (*m*/*z*)	Retention Time (min)	Charge	Fragmentor (V)	Quantitative Product Ion (*m*/*z*)	Collision Energy (eV)	Qualitative Product Ion (*m*/*z*)	Collision Energy (eV)	Species
1	766.6	6.8	3	140	535.2	30	478.2	30	Donkey
2	761.4	7.2	3	160	598.3	20	478.2	30	Donkey
3	469.3	2.8	2	110	712.3	10	783.4	15	Donkey
4	618.4	2.8	2	130	779.4	20	850.4	20	Donkey
5	386.3	2.0	2	90	499.3	3	556.3	5	Horse
6	604.8	4.8	2	130	910.5	25	570.3	20	Cattle
7	596.8	5.3	2	140	447.8	15	570.3	20	Cattle
8	490.5	3.2	2	80	566.2	5	807.5	10	Pig
9	773.9	5.0	2	180	977.6	30	556.3	35	Pig

^a^ The quantitative ion pairs are (1) donkey-specific marker 4, 766.7 > 535.2; (2) donkey-specific marker 3, 761.4 > 598.3; (3) donkey marker in the Chinese Pharmacopoeia, 469.3 > 712.3; (4) donkey marker in the Chinese Pharmacopoeia, 618.4 > 779.4; (5) horse-specific marker 1, 386.3 > 499.3; (6) cattle-specific marker 2, 604.8 > 910.5; (7) cattle-specific marker 1, 596.8 > 447.8; (8) pig-specific marker 2, 490.5 > 566.2; (9) pig-specific marker 5, 773.9 > 977.6. The ion pairs are detected by UPLC-QQQ-MS/MS in MRM mode.

**Table 5 molecules-27-04643-t005:** Validation of the established quantitative method using four individual species-specific markers (DM4, HM1, CM2, and PM2).

Marker No.	Linear Regression ^a^	R2	LOD ^b^ (mg)	LOQ ^b^ (mg)	Analyte	Repeatability		Recovery % (RSD, *n* = 3)		
						Intra-Day (*n* = 6)	Inter-Day (*n* = 3)	Low	Middle	High
donkey-specific marker 4 DM4	*y* = 44.607*x* + 49.776	0.9992	0.05	0.16	Home-made donkey-hide gelatin	2.3%	3.9%	86 (3.3%)	92 (3.1%)	89 (4.5%)
					Commercial Ejiao sample	1.6%	3.8%	88 (4.7%)	89 (3.9%)	93 (4.3%)
horse-specific marker 1 HM1	*y* = 124.66*x* + 151.13	0.9961	0.02	0.08	Home-made horse-hide gelatin	0.7%	2.5%	97 (1.1%)	99 (2.3%)	102 (1.4%)
					Commercial Ejiao sample	0.8%	2.3%	98 (2.5%)	105 (1.7%)	114 (1.6%)
cattle-specific marker 2 CM2	*y* = 106.14*x* + 157.72	0.9927	0.02	0.08	Home-made cattle-hide gelatin	1.2%	3.3%	89 (3.5%)	93 (3.7%)	99 (4.1%)
					Commercial Ejiao sample	1.3%	2.8%	96 (4.2%)	87 (3.3%)	109 (3.5%)
pig-specific marker 2 PM2	*y* = 299.07*x* + 162.53	0.9991	0.03	0.09	Home-made pig-hide gelatin	0.9%	3.9%	104 (4.1%)	112 (3.7%)	118 (3.2%)
					Commercial Ejiao sample	1.7%	4.1%	99 (3.0%)	103 (4.9%)	112 (4.5%)

^a^ The regression was plotted by standard material amount vs. marker peak area; ^b^ Limits of detection (LOD) and limits of quantitation (LOQ) of standard material under the present conditions were calculated using the following formulas. LOD was calculated using standard solutions’ signal-to-noise ratios (S/N) using the definition S/N > 3. LOQ was calculated from standard solutions’ signal-to-noise ratios (S/N) using the definition S/N > 10. The result is based on specific markers including donkey-specific marker (DM4, *m*/*z* 766.6952; MRM, 766.7 > 535.2), horse-specific marker (HM1, *m*/*z* 386.2108; MRM, 386.3 > 499.3), cattle-specific marker (CM2, *m*/*z* 604.8556; MRM, 604.8 > 910.5), and pig-specific marker (PM2, *m*/*z* 490.5942; MRM, 490.5 > 566.2).

**Table 6 molecules-27-04643-t006:** Quantitative results of commercial donkey-hide gelatin samples based on our marker and the markers in the Chinese Pharmacopoeia.

Sample No.	Result (%) mg/mg						
	Donkey (CP) ^a^	Donkey (DM4) ^a^	Horse (HM1)	Cattle (CM2)	Pig (PM2)	Donkey (CP) ^b^	Identification
1	54.0	62.3	/ ^c^	/	/	NA ^c^	NA
2	56.1	80.1	/	/	/	NA	NA
3	60.5	84.2	/	/	/	NA	NA
4	55.3	57.8	/	/	/	NA	NA
5	0.2	/	0.2	52.3	/	NA	NA
6	/	/	/	59.9	/	NA	NA
7	/	/	/	43.5	/	NA	NA
8	0.2	/	1.6	6.2	0.0	NA	NA
9	61.4	89.4	/	/	/	NA	NA
10	57.3	91.2	/	/	/	NA	NA
11	74.2	66.3	/	/	/	NA	NA
12	53.3	57.4	/	/	/	NA	NA
13	70.9	69.9	/	/	/	NA	NA
14	/	/	/	54.3	/	NA	NA
15	/	/	/	39.0	/	NA	NA
16	75.7	77.3	/	/	/	NA	NA
17	66.0	78.8	/	/	/	NA	NA
18	71.5	67.1	/	/	/	NA	NA
19	/	/	/	49.0	/	NA	NA
20	/	/	/	29.0	/	NA	NA
21	/	/	/	47.4	/	NA	NA
22	/	/	/	63.3	/	NA	NA
23	/	/	/	45.5	/	NA	NA
24	66.5	69.0	/	/	/	NA	NA
25	76.2	70.4	/	/	/	NA	NA
26	69.5	57.0	/	/	/	NA	NA
27	12.6	2.1	18.5	/	47.4	2.8	✓ ^c^
28	/	/	/	49.3	/	NA	NA
29	/	/	/	40.2	0.1	NA	NA
30	/	/	/	67.4	/	NA	NA
31	66.8	74.5	/	/	/	NA	NA
32	66.7	72.9	/	/	/	NA	NA
33	0.5	/	0.6	31.5	/	NA	NA
34	/	/	/	46.3	/	NA	NA
35	48.6	78.9	/	/	/	NA	NA
36	70.8	63.3	4.1	/	/	NA	NA
37	19.3	/	26.3	/	33.5	5.5	× ^c^
38	/	/	/	73.4	/	NA	NA
39	59.9	91.9	/	/	/	NA	NA
40	44.3	70.2	/	/	/	NA	NA
41	72.1	64.6	/	/	/	NA	NA
42	/	/	/	42.2	/	NA	NA
43	/	/	/	44.9	/	NA	NA
44	31.9	6.4	40.4	38.7	/	11.2	✓
45	/	/	/	47.8	/	NA	NA
46	70.1	66.5	/	/	/	NA	NA
47	70.7	57.4	/	/	/	NA	NA
48	98.3	98.8	/	/	/	NA	NA
49	/	/	/	47.0	/	NA	NA
50	69.9	88.0	/	/	/	NA	NA
51	13.4	/	21.4	/	42.7	2.2	✓
52	96.3	96.1	/	/	/	NA	NA
53	/	/	/	44.4	/	NA	NA
54	68.2	55.5	/	/	/	NA	NA
55	50.8	61.8	/	/	/	NA	NA
56	74.8	70.7	/	/	/	NA	NA
57	53.7	72.1	/	/	/	NA	NA
58	/	/	/	46.9	/	NA	NA
59	/	/	/	65.7	/	NA	NA
60	/	/	/	82.9	/	NA	NA
61	/	/	/	62.7	/	NA	NA
62	80.5	69.6	/	/	/	NA	NA
63	51.4	71.7	/	/	/	NA	NA
64	/	/	/	70.8	/	NA	NA
65	/	/	/	56.3	/	NA	NA
66	/	/	/	69.4	/	NA	NA
67	72.7	71.9	/	/	/	NA	NA
68	79.9	75.9	/	/	/	NA	NA
69	/	/	/	53.0	/	NA	NA
70	/	/	/	59.3	/	NA	NA
71	78.5	/	107.7	/	/	14.7	×
72	70.8	77.8	/	/	/	NA	NA
73	/	/	/	41.7	/	NA	NA
74	61.2	70.9	/	/	/	NA	NA
75	53.8	60.3	/	/	/	NA	NA
76	50.2	71.8	/	/	/	NA	NA
77	/	/	0.3	54.1	/	NA	NA
78	/	/	/	45.7	/	NA	NA
79	/	/	/	60.3	/	NA	NA
80	/	/	/	65.0	/	NA	NA
81	83.9	73.8	/	/	/	NA	NA
82	/	/	/	71.0	/	NA	NA
83	57.6	87.9	/	/	/	NA	NA
84	85.3	76.2	/	/	/	NA	NA
85	77.8	71.4	/	/	/	NA	NA
86	53.2	57.6	/	/	/	NA	NA
87	/	/	/	52.4	/	NA	NA
88	/	/	/	29.1	/	NA	NA
89	/	/	/	47.9	/	NA	NA
90	/	/	/	43.7	/	NA	NA
91	75.0	68.2	/	/	/	NA	NA
92	80.6	68.2	/	/	/	NA	NA
93	59.9	56.3	/	/	/	NA	NA
94	67.7	74.7	/	/	/	NA	NA
95	0.9	/	1.3	47.5	/	NA	NA
96	/	/	/	40.9	/	NA	NA
97	/	/	/	46.5	/	NA	NA
98	/	/	/	44.3	/	NA	NA
99	69.7	53.4	/	/	/	NA	NA
100	79.6	5.7	105.9	/	/	26.2	✓
101	71.8	6.3	99.3	/	/	21.8	✓
102	60.9	71.4	/	/	/	NA	NA
103	52.4	/	78.5	/	/	12.7	×
104	34.3	/	46.5	/	/	10.5	×
105	63.2	3.9	107.8	/	/	8.9	×
106	3.0	/	24.1	/	/	/	✓
107	93.6	92.8	/	/	/	NA	NA
108	/	/	/	/	/	NA	NA
109	62.8	90.4	/	/	/	NA	NA
110	20.2	32.6	/	/	/	NA	NA

^a^ The content is directly calculated with peak area of the specific marker. ^b^ The content is indirectly calculated by peak area of the specific marker by subtracting the corresponding peak area from horse-hide gelatin. ^c^ “✓” indicates that the authentication result by indirect calculation is true. “×” means that the authentication result by indirect calculation is false. If the content is not more than 5%, it indicates that the marker in the species was not detected. “/” means that the marker from the animal was not detected by UPLC-QQQ-MS/MS. “NA” indicates “not applicable”. Abbreviation: “Donkey (CP)” represents the marker in the Chinese Pharmacopoeia, 469.3 > 712.3. “Donkey (DM4)” represents donkey-specific marker 4, 766.7 > 535.2. “Horse (HM1)” represents horse-specific marker 1, 386.3 > 499.3. “Cattle (CM2)” represents cattle-specific marker 2, 604.8 > 910.5. “Pig (PM2)” represents pig-specific marker 2, 490.5 > 566.2.

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
