# Peer review of "Qualitative and Quantitative Analysis of Ejiao-Related Animal Gelatins through Peptide Markers Using LC-QTOF-MS/MS and Scheduled Multiple Reaction Monitoring (MRM) by LC-QQQ-MS/MS"

_molecules, 2022, doi:10.3390/molecules27144643_

Round 1
Reviewer 1 Report
In the manuscript entitled “ Qualitative and quantitative analysis of Ejiao related animal gelatins through peptide markers using LC-QTOF-MS/MS and scheduled multiple reaction monitoring (MRM) by LC-QQQ-MS/MS” Authors analyzed gelatins from 4 species (donkey, horse, cattle, pig) to determine specific peptides for donkey gelatin identification. For 4 from 7 new donkey-specific marker peptides ions, had sufficient score in de novo sequencing software (PEAKS Studio)-above 50%. Targeted quantitative analysis demonstrated the applicability of the peptides detected by the authors to confirm the identity of donkey gelatin.
The manuscript is well structured and provides a thorough authentication analysis of gelatin from different species. In my opinion, it is suitable for publication in Molecules.
Author Response
In the manuscript entitled “ Qualitative and quantitative analysis of Ejiao related animal gelatins through peptide markers using LC-QTOF-MS/MS and scheduled multiple reaction monitoring (MRM) by LC-QQQ-MS/MS” Authors analyzed gelatins from 4 species (donkey, horse, cattle, pig) to determine specific peptides for donkey gelatin identification. For 4 from 7 new donkey-specific marker peptides ions, had sufficient score in de novo sequencing software (PEAKS Studio)-above 50%. Targeted quantitative analysis demonstrated the applicability of the peptides detected by the authors to confirm the identity of donkey gelatin.
The manuscript is well structured and provides a thorough authentication analysis of gelatin from different species. In my opinion, it is suitable for publication in Molecules.
Authors’ Response------ Thank you very much for your positive comments!

Reviewer 2 Report
A definitely interesting article reporing the discovery of specific markers that can held authenticate products related to traditional chinese medicine. The article is - in general terms - well written and rather easy to folow. The level of scientific English is also acceptable by me. The research does fall winthin the scope of Molecules and I suggest acceptance after minor revision.
Keywords: a couple of additional keywords shouls be added by the authors to ensure a more efficient bibliographic coverage.
Introduction: the introductory section is somewhat short; more information should be included by the authors (e.g. current research and legislation on the specific topic).
Section 2.4: In the first paragraph please replace "FIrst, Second, Third ...." with (i), (ii), (iii) .....
Section 3.2: A reference would be potentialy useful (sample processing).
Author Response
A definitely interesting article reporing the discovery of specific markers that can held authenticate products related to traditional chinese medicine. The article is - in general terms - well written and rather easy to folow. The level of scientific English is also acceptable by me. The research does fall winthin the scope of Molecules and I suggest acceptance after minor revision.
Authors’ Response------ Thank you very much for your positive comments!
Keywords: a couple of additional keywords shouls be added by the authors to ensure a more efficient bibliographic coverage.
Authors’ Response------ Thanks for the valuable comment and suggestion. We have added another two keywords (database-independent; de novo sequencing).
Introduction: the introductory section is somewhat short; more information should be included by the authors (e.g. current research and legislation on the specific topic).
Authors’ Response------ Thanks for the valuable comment and suggestion. We have added more information in the introduction section (lines 48-69:” Peptide markers have been developed for authentication purposes. For example, in the Chinese Pharmacopoeia, three peptides are used as authentication markers for Ejiao [2]. Li, et al. found two peptide markers for Ejiao by an improved proteomics approach, combining bioinformatics-based prediction [8]. Liu, et al. found two peptide markers for deer-hide gelatin by combining set theory analysis. The result suggested that bioinformatics-based prediction might ignore posttranslational modifications that occurred during the extraction procedure, which will cause a false-positive [9].
The specificity of these markers relies on the proteomics database (Fig. 1A) [2, 8-15], but many of the databases are shared by different species, e.g. that shared by donkey (Equus asinus) and horse (Equus caballus) [16-19]. Kumazawa, et al. identified a total of 396 peptides for deer glue, including 55 peptides from bovine (Bos taurus) and 37 peptides from sheep (Ovis aries) [16]. The myoglobin peptide HPGDFGADAQGAMTK is horse-specific in the Uniprot database [20]; however, it also exists in donkeys [18]. Furthermore, Prandi, et al. found all the peptides identified in the horse meat were also present in the donkey meat. Therefore, this myoglobin peptide has to be used as a marker peptide for equine meat (horse and donkey) in their study [18]. The specific peptide GPPGAAGPPGPR for authentication of Ejiao in the Chinese Pharmacopoeia is also found in horse-hide gelatin [2, 8]. More importantly, many peptides are not recorded in these protein databases [16, 18]. For example, information on pig COL1A1 is lacking [8]. The deer type I collagen sequence is not included in public databases [16].”).
Section 2.4: In the first paragraph please replace "FIrst, Second, Third ...." with (i), (ii), (iii) .....
Authors’ Response------ Thanks! According to your suggestion, the related sentences in Section 2.4 have been revised.
Section 3.2: A reference would be potentialy useful (sample processing).
Authors’ Response------ Thanks! According to your suggestion, a reference has been added in Section 3.2 (lines 283-284: “The sample pretreatment process was based on our previous study with minor modification [28].”).
